# Aqueous Extracts from Hemp Seeds as a New Weapon against *Staphylococcus epidermidis* Biofilms

**DOI:** 10.3390/ijms242216026

**Published:** 2023-11-07

**Authors:** Angela Casillo, Caterina D’Angelo, Paola Imbimbo, Daria Maria Monti, Ermenegilda Parrilli, Rosa Lanzetta, Giovanna Gomez d’Ayala, Salvatore Mallardo, Maria Michela Corsaro, Donatella Duraccio

**Affiliations:** 1Department of Chemical Sciences, University of Naples “Federico II”, Via Cintia 21, 80126 Napoli, Italy; angela.casillo@unina.it (A.C.); caterina.dangelo@unina.it (C.D.); paola.imbimbo@unina.it (P.I.); mdmonti@unina.it (D.M.M.); erparril@unina.it (E.P.); lanzetta@unina.it (R.L.); corsaro@unina.it (M.M.C.); 2Institute of Polymers, Composites and Biomaterials (IPCB)-CNR, Via Campi Flegrei 34, 80078 Pozzuoli, Italy; salvatore.mallardo@ipcb.cnr.it; 3Institute of Sciences and Technologies for Sustainable Energy and Mobility (STEMS)-CNR, Strada Delle Cacce 73, 10135 Torino, Italy; donatella.duraccio@stems.cnr.it

**Keywords:** *Cannabis sativa* L., seeds, hemp extract, antibiofilm activity, *Staphylococcus epidermidis*

## Abstract

This study investigated the antibiofilm activity of water-soluble extracts obtained under different pH conditions from *Cannabis sativa* seeds and from previously defatted seeds. The chemical composition of the extracts, determined through GC-MS and NMR, revealed complex mixtures of fatty acids, monosaccharides, amino acids and glycerol in ratios depending on extraction pH. In particular, the extract obtained at pH 7 from defatted seeds (Ex7d) contained a larger variety of sugars compared to the others. Saturated and unsaturated fatty acids were found in all of the analysed extracts, but linoleic acid (C18:2) was detected only in the extracts obtained at pH 7 and pH 10. The extracts did not show cytotoxicity to HaCaT cells and significantly inhibited the formation of *Staphylococcus epidermidis* biofilms. The exception was the extract obtained at pH 10, which appeared to be less active. Ex7d showed the highest antibiofilm activity, i.e., around 90%. Ex7d was further fractionated by HPLC, and the antibiofilm activity of all fractions was evaluated. The 2D-NMR analysis highlighted that the most active fraction was largely composed of glycerolipids. This evidence suggested that these molecules are probably responsible for the observed antibiofilm effect but does not exclude a possible synergistic contribution by the other components.

## 1. Introduction

Hemp (*Cannabis sativa* L.) is an annual plant belonging to the Cannabaceae family that is widespread thanks to its capability to adapt to different climate conditions. It is grown and used for a wide variety of purposes, such as building material, textile fibre, paper and fuel. Due to its edible oil and high-quality fibre, hemp has been extensively used for food and as a fibre source [1,2,3,4]. Moreover, it contains several secondary metabolites, such as terpenoids, phytocannabinoids and flavonoids, which can be used for different medical treatments thanks to their anti-inflammatory, antioxidative and antimicrobial activity [5,6,7]. Many studies suggest that essential oils extracted from different varieties of *Cannabis sativa L.* exhibit antibacterial and antibiofilm effects against Gram-positive and Gram-negative bacteria [8,9,10,11]. Krejčí et al. first identified the compound with antibiotic activity and named it cannabidiolic acid (CBDA) [12]. Since that time, other hemp components with antibiotic and antibiofilm activities have been found and characterized [13,14,15].

*Cannabis sativa* L. hemp seeds are also highly valued for their components with nutritional properties and promising health benefits [16]. They contain 25–35% lipids, 20–25% proteins and 30% carbohydrates, thus representing a reliable source of nutrients and non-nutrients [17]. They are mainly employed in animal feed, but their extracts contain products like oil and protein powder that are particularly promising for applications in human nutrition [1,18]. It has been demonstrated that hemp seed shows a significant therapeutic effect against several diseases [19] and can be used for food as a source of antioxidant compounds [20]. Although oils are the most-studied and most-characterised hemp-seed extracts due to their well-known antioxidant, anticancer and antimicrobial properties [16,21,22], the hydrophilic component is equally promising. It has been demonstrated that hemp seeds’ water-soluble fraction exhibits particularly intriguing properties, such as anti-hypersensitivity, anti-inflammatory, and antioxidant potential [23,24,25,26]. Interest in these extracts has even more economic potential because they can be obtained from waste seeds that for various reasons (e.g., defects, aging) are considered not suitable for the food/agriculture market. Therefore, the use of hemp seeds as a source of bioactive molecules is particularly promising from a circular bioeconomy perspective.

The purpose of this study was to explore the potential antibiofilm activity of water-soluble extracts from *Cannabis sativa* L. seeds and to identify the bioactive metabolites through an activity-guided purification approach. It is to be noted that the discovery of most of natural products has been driven by bioactivity-guided fractionation of chemical extracts [27]. Currently, although new strategies have been introduced [28,29], this approach remains effective [30,31,32,33,34]. To this end, aqueous extracts obtained under different pH conditions from both untreated and defatted seeds were investigated for their antibiofilm effect against *Staphylococcus epidermidis*.

Although *S. epidermidis* is a harmless colonizer of human skin, it has emerged as an important opportunistic pathogen in infections associated with medical devices [35], causing approximately 30% to 43% of joint-prosthesis infections [36,37] and fracture-fixation infections [38]. *S. epidermidis*’s remarkable ability to form biofilms is widely regarded as its major pathogenic determinant [39]. Indeed, the success of *S. epidermidis* as a pathogen is attributed to its ability to migrate from the skin, along the surface of the device and into the body, where it can adhere to both biotic and abiotic surfaces, forming a highly organised bacterial community known as a biofilm. A biofilm is a microbially-derived sessile community characterised by cells attached either to a substratum or to an interface and embedded in a self-produced matrix of extracellular polymeric substance [40]. Under these conditions, bacteria are embedded and protected from external assaults, and they are more resistant to conventional antibiotic treatments than are planktonic bacteria, thus resulting in recalcitrant biofilm-associated infections [41]. Indeed, it is well known that biofilm-forming bacteria are about one thousand times more resistant to antimicrobials than are planktonic cells [42]. Considering this problem, the prevention of biofilm formation and the treatment of existing biofilms is currently a difficult challenge, and the discovery of new multi-targeted or combinatorial therapies is increasingly urgent [41,43].

To the best of our knowledge, no studies on the antibiofilm activity of aqueous extracts from hemp seeds are reported in the literature, although various examples of bioactive oils are described. For example, Frassinetti et al. demonstrated that a seed extract in ethanol has selective inhibitory activity against an *S. aureus* strain, as well as potential as a new antibiofilm agent [44]. Moreover, Menjivar et al. demonstrated that hempseed oil had the most effective anti-biofilm properties against *P. aeruginosa* and *S. epidermidis* [45].

To overcome the negative effects related to the use of organic solvents such as elevated costs and the risk of contamination associated with extraction, we investigated the antibiofilm effect of aqueous seed extracts against *S. epidermidis*. The extraction was performed in water under different pH conditions, and extracts were chemically characterised to determine their composition. The investigations were performed using GC-MS [46,47,48,49,50], HPLC [51,52,53,54,55] and NMR spectroscopy [56,57,58]. Finally, the biocompatibility of the extracts was evaluated in a cell-based model.

## 2. Results and Discussion

Polar components of hemp seeds were extracted through a very sustainable procedure under different pH conditions. The extraction yield from hemp seeds was 30.27 ± 0.03% at pH 5, 35.13 ± 0.11% at pH 7 and 25.28 ± 0.23% at pH 10. For defatted seeds, the yield of the extraction carried out at pH 7 was 34.46 ± 0.15%.

All of the extracts (Exs) were revealed to be complex mixtures on analysis by ^1^H-NMR spectroscopy (Figure 1).

The signals in the region between 3–5 ppm in the ^1^H-NMR spectra suggest the presence of carbohydrates, whereas those between 0.5–3 ppm could be attributable to aliphatic carbon chains. The signals in the range of 6–8 ppm can be ascribed to the presence of peptides or amino acids. The NMR spectra suggest that aliphatic proton signals and the signals attributable to amino acids were more abundant in Ex7d and Ex4.5 compared to the other two extracts. By contrast, in Ex7 and Ex10, the carbinolic proton signals were more pronounced. Finally, Ex7d and Ex4.5 spectra showed signals between 6 and 8 ppm that are attributable to aromatic rings, whereas the same signals in Ex7 and Ex10 were present at a very low intensity.

All of the extracts were subjected to chemical derivatization to convert all of the compounds into volatile derivatives. They were then analysed by GC-MS. The results of chemical analyses (Figure 2 and Figure 3, Table 1 and Table 2) agreed with the results of NMR analysis. Indeed, all of the extracts were shown to contain a mixture of fatty acids, amino acids, and monosaccharides. Ex7d and Ex4.5 are composed mainly of amino acids and saturated fatty acids, while Ex7 and Ex10 are composed mainly of glycerol, the sugars glucose and galactose and unsaturated fatty acids. The compounds were identified by comparing retention times in both the TIC and the mass spectrum with those of authentic standards, which were also derivatised with the same procedure. As an example, Appendix A shows the chromatograms and the mass spectra of both Ex4.5 and galactose and the galacturonic acid standard used for its identification. All of the components of each extract were identified in the same way. The absence of cannabidiol and tetrahydrocannabinol peaks from these chromatograms could be due to the high molecular weight of these compounds after derivatization [59]. In fact, even when the final temperature of the GC program was 300 °C, the library did not detect any of these components. On the other hand, the comparison of our results with those obtained using other aqueous extracts from hemp seeds suggested no other differences [59].

To evaluate the effect of extracts on bacterial viability, the dried Exs, diluted to the desired final concentration in BHI medium, were analysed to investigate their antimicrobial activity against *S. epidermidis* RP62A.

To this end, an appropriate dilution (0.125 OD600nm) of *S. epidermidis* RP62A bacterial culture in the exponential phase was used. No antimicrobial activity against *S. epidermidis* RP62A strains was observed for any tested Exs (maximum concentration tested 500 µg mL^−1^) (Appendix A).

The antibiofilm effect of the extracts at a concentration of 1 mg mL^−1^ was examined against two staphylococcal strains: *S. epidermidis* O-47 [60,61] and *S. epidermidis* RP62A [62]. The *S. epidermidis* O-47 strain is methicillin-susceptible, and it was isolated from patients with orthopaedic implant infections [63]. It is a naturally occurring *agr* mutant [61] and produces only very low levels of delta-toxin and other phenol-soluble modulins (PSMs), the transcription of which is agr-controlled [61,64]. As previously reported [65], the *agr*-negative genotype of *S. epidermidis* O-47 is responsible for increased expression of the surface protein AtlE, a bifunctional adhesin/autolysin abundant in the cell wall of *S. epidermidis*, and this overexpression enhances biofilm formation on polymer surfaces. *S. epidermidis* RP62A (ATCC 35984) is a methicillin-resistant strain [66] isolated from an outbreak of intravascular catheter-associated sepsis [67].

As shown in Figure 4, almost all of the extracts were able to inhibit biofilm formation by both staphylococcal strains. The only exception was Ex10, which was less active. It is interesting to note that Ex7d, Ex7 and Ex4.5 had similar activity profiles against the two target strains; they are all more effective on the *S. epidermidis* O-47 biofilm than on the *S. epidermidis* RP62A biofilm. The two *S. epidermidis* strains differ in several aspects [60]. One of the reported differences between *S. epidermidis* RP62A and *S. epidermidis* O-47 is the amount of AtlE present in the cell envelope [61]. Future studies will clarify whether this characteristic is involved in the difference in the extracts’ efficacy. In any case, the phenotypic and genotypic heterogeneity of *S. epidermidis* strains [68,69,70] could limit the clinical applicability of extracts. Therefore, future studies will be dedicated to testing the extracts against different *S. epidermidis* strains.

The observation that the extracts have similar activity profiles against the two target strains could suggest that in the three extracts, the molecule/s involved in antibiofilm activity is/are the same and that the slight difference in activity is related to the concentration of the compound/s. It is difficult to ascribe this behaviour to specific key component/s, as the antibiofilm effect of such complex mixtures can depend on interactions (enhancing and/or inhibiting the bioactivity) between the components.

Because all of the extracts were active at a concentration of 1 mg mL^−1^, their anti-biofilm effect against staphylococcal strains was evaluated at different concentrations within the 0.5–0.01 mg mL^−1^ range. As reported in Figure 5, the effect was concentration-dependent, and the activity of the extracts decreased as their concentrations decreased. Ex7d showed the highest activity against both target strains, even at the lowest tested concentration (0.01 mg mL^−1^).

In order to characterize the compositions of the extracts that exhibited antibiofilm activity (Ex7d, Ex7 and Ex4.5), the protein content was evaluated by BCA assay and SDS-PAGE followed by Comassie-blue staining. No proteins were present in any of the tested extracts, as shown in Appendix A. The absence of proteins in the tested extracts agrees with findings in the literature, in which it is reported that degreasing or defatting is a fundamental step in extracting proteins from plant seeds, as the presence of oils in general causes the formation of lipid-protein complexes, reducing the extraction yield [71]. Nonetheless, no proteins were found even in Ex7b, which was obtained from defatted hemp seeds, because the extraction pH (pH 7) was not optimal for dissolving hydrophilic proteins [71].

The biocompatibility of the extracts that exhibited antibiofilm activity (Ex7d, Ex7 and Ex4.5) was evaluated by using immortalised human keratinocytes to assess the potential applicability of these extracts to humans. HaCaT cells were incubated with increasing concentrations of each extract (i.e., from 0.1 to 100 µg mL^−1^), and their viability was evaluated at 24 h and 48 h. Figure 6A,B shows that none of the analysed extracts significantly affect cell viability at the investigated concentrations at 24 and 48 h, respectively. Only Ex4.5 slightly reduced cell viability after 24 h at a concentration of 100 µg mL^−1^ and after 48 h at a concentration of 50 µg mL^−1^. The absence of toxic effects makes the extracts suitable for the intended application.

According to an activity-guided approach, Ex7d was further fractionated by HPLC on a gel filtration column because it exhibited the highest antibiofilm activity against the two *S. epidermidis* strains (Appendix A). This strategy aims to evaluate the antibiofilm effect of the separated fractions and thus to potentially identify the component/s responsible for the observed activity. The obtained fractions (A–E) were tested against both *S. epidermidis* strains at a concentration of 0.1 mg mL^−1^ (Figure 7).

Fraction B proved to be the most effective in countering biofilm formation, especially against the O-47 strain, the biofilm production of which was reduced by 90%. The other fractions induced a reduction in biofilm occurrence in both strains that ranged between 20% and 60%. Fraction B was therefore subjected to NMR analysis. The 2D-NMR spectra suggest that this fraction is still a mixture of different components.

The 2D-NMR results shown in Figure 8 suggest that the purified fraction responsible for the antibiofilm activity is mainly composed of fatty-acid chains, glycerol, and alditols. Alditols were revealed here for the first time, likely because their concentration in the unpurified extract was not high enough for them to have been detected before. In particular, the presence of the -CH_2_OH proton signals around δ 3.6–3.8 ppm in correlation with -CH_2_OH carbon signals at δ 62 ppm and -CHOH indicate the presence of glycerol. Moreover, the additional -CHOH signals in the range 3.8–4.5 ppm for ^1^H nucleus and in the 70–80 ppm region for ^13^C signals suggest the presence of inositol. Finally, the numerous -CH_2_- signals may indicate the presence of methylene groups of fatty acids.

Until now, researchers have primarily investigated organic extracts from hemp seeds for antibiofilm activity. A bioactive ethanol extract from hemp seeds was analysed by mass spectrometry, and free fatty acids and phospho-glycerol derivatives, as well as flavanols and amino acids, were found [72]. According to these results, it can be hypothesised that the antibiofilm activity is mainly attributable to glycerolipids, although it is not possible to exclude the possibility of synergy with the other components.

## 3. Materials and Methods

### 3.1. Extraction of Polar Components from Hemp Seeds

The water-soluble components of hemp seeds were extracted according to the procedure described by Wen et al. [19]. Three extractions from dry seeds were performed at different pH values: acidic (pH 4.5, Ex4.5), neutral (pH 7.0, Ex7) and alkaline (pH 10.2, Ex1), as it is widely demonstrated that the quantities and types of a substance extracted depend on the extraction conditions, i.e., solvent and pH) [73,74,75]. Dried and ground hemp seeds were soaked in the aqueous extraction buffer at 60 °C and subjected to magnetic agitation for 3 h. Afterwards, supernatants were collected by centrifugation for 10 min at 8000 rpm, then concentrated and dried. In a parallel experiment, an extraction under neutral pH conditions was carried out on previously defatted seeds obtained by Soxhlet extraction. Briefly, a filter-paper extraction thimble was filled with 10 g of ground hemp seeds and the extraction was carried out with 150 mL of n-hexane for 3 h at 70 °C. Once the defatted seeds were obtained, the extraction was performed in water at 60 °C, as previously described (Ex7d).

### 3.2. Protein Determination

The protein concentration of the extracts was determined using the BCA Assay Kit (Thermo Fisher Scientific, Waltham, MA, USA) and by SDS-PAGE analyses followed by Comassie-blue staining.

### 3.3. GC-MS Analyses

Dried samples were analysed by gas chromatography-mass spectrometry (GC-MS) after derivatisation [76].

The extracts (2 mg) were hydrolysed with 6 M HCl (0.1 mL) at 110 °C for 16 h. Then, the samples were treated at 80 °C for 90 min with 0.5 M of HCl/CH_3_OH. Both treatments were used to obtain monomers to be derivatised as acetylated compounds. The methodology allowed the isolation of mixtures of fatty acid methyl esters (FAMEs), *O*-methyl glycosides, and methyl esters of amino acids. The fatty acids were separated from the mixtures through three extractions with hexane, then dried and analysed by GC-MS [77]. The remaining methanol layers were dried, acetylated with pyridine 50 µL and acetic anhydride 50 µL (100 °C, 30 min), dissolved in acetone and injected.

All the samples were analysed with the GC-MS Agilent Technologies 7820A, which was equipped with a mass selective detector, 5977B (HP-5 capillary column; 30 m × 0.25 mm i.d.; flow rate, 1 mL min^−1^; He as carrier gas), adopting the following temperature program: 90 °C for 3 min, from 90 to 300 °C at 15 °C/min and 300 °C for 5 min. The run time was 23 min. All the compounds were recognised by comparing their mass spectra and retention times with those of authentic standards. In addition, the library (NIST MS Search 2.3) of the GC-MS confirmed the correctness of the structure attribution.

### 3.4. NMR Spectroscopy

^1^H and ^1^H-^13^C distortionless enhancement by polarisation transfer-heteronuclear single quantum coherence (DEPT-HSQC) were acquired using a Bruker Avance 600 MHz spectrometer equipped with a Cryoprobe at 298 K in D_2_O. All the experiments were performed using standard pulse sequences available in the Bruker software, as already reported [78,79].

### 3.5. Gel Filtration Chromatography

Extract 3 was further fractionated by HPLC on a TSK-5000 gel filtration column, eluted with water, and associated with a refractive index. The collected fractions were freeze-dried.

### 3.6. Bacterial Strains and Culture Conditions

The bacterial strains used in this work were *S. epidermidis* O-47, a strain isolated from a case of clinical septic arthritis and kindly provided by Prof. Gotz [80], and *S. epidermidis* RP62A, a reference strain isolated from an infected catheter (ATCC collection no. 35984) [67,81].

Staphylococci were grown at 37 °C in Brain Heart Infusion broth (BHI, Oxoid, UK); biofilm formation was assessed in static conditions, while planktonic cultures were grown under agitation (180 rpm).

### 3.7. Determination of Minimal Inhibitory Concentrations 

The MIC of the extracts was defined as the lowest concentration at which observable bacterial growth was inhibited. It was determined for *S. epidermidis* RP62A according to the guidelines of the Clinical Laboratory Standards Institute (CLSI 2018). Briefly, an *S. epidermidis* RP62A overnight culture was diluted at 0.5 McFarland (0.125 OD600 nm), and 90 μL of this dilution was added to each well of a 96-well microtiter dish. Next, 10 µL of extracts previously diluted in BHI to the desired final concentration was added to each well. Then, 10 µL BHI was added to the control lane (CN). The microtiter plate was incubated at 37 °C for 24 h. After incubation, the MIC was determined. All experiments were performed in quadruplicate.

### 3.8. Antibiofilm Assay

The quantification of in vitro biofilm production was based on the method described by Christensen, with slight modifications [82]. For staphylococcal biofilm formation in the presence of extracts or HPLC fractions, the wells of a sterile 96-well flat-bottomed polystyrene plate were filled with *S. epidermidis* RP62A or *S. epidermidis* O-47 cultures in the exponential growth phase diluted in BHI to final concentration of about 0.1 and 0.001 OD600 nm, respectively. The dried extracts or HPLC fractions were dissolved in the culture medium BHI. Each well was filled with 100 μL of culture and 100 μL of extract (or fraction) at the desired final concentration. As a control (untreated bacteria), the first row was filled with 100 μL of culture and 100 μL of BHI. The plates were incubated aerobically for 24 h at 37 °C. Biofilm formation was measured using crystal violet staining. After incubation, planktonic cells were gently removed and wells were washed three times with double-distilled water and thoroughly dried. Each well was then stained with 0.1% crystal violet, incubated for 15 min at room temperature, rinsed twice with double-distilled water and thoroughly dried. The dye bound to adherent cells was solubilised with 20% (*v*/*v*) glacial acetic acid and 80% (*v*/*v*) ethanol. After 30 min of incubation at room temperature, the total biofilm biomass in each well was spectrophotometrically quantified at 590 nm.

### 3.9. Cytotoxicity Assay

Immortalized human keratinocytes (HaCaT, Innoprot, Derio, Spain) were cultured in Dulbecco’s modified Eagle’s medium (DMEM) (Sigma-Aldrich, St. Louis, MO, USA) supplemented with 10% foetal bovine serum (HyClone, Logan, UT, USA), 2 mM L-glutamine and antibiotics. Cells were grown in a 5% CO_2_ humidified atmosphere at 37 °C and seeded in 96-well plates at a density of 2 × 10^3^ cells/well. After 24 h, cells were incubated with increasing concentrations of each extract (0.1–100 µg mL^−1^) for 24 and 48 h. At the end of incubation, cell viability was assessed by the MTT assay, as previously reported [83]. Cell viability was expressed as the percentage of viable cells in the presence of each extract compared to the controls, namely, untreated cells and cells supplemented with identical volumes of PBS. Each sample was tested in three independent analyses, each of which was carried out in triplicate.

## 4. Conclusions

The problem of drug-resistant bacterial strains and the production of biofilms is a significant global health concern that pushed us to search for alternative solutions. In recent decades, the use of plants as a source of bioactive molecules represents one of the most promising approaches. Among these plant products, hemp seeds are particularly interesting due to their high content of small active metabolites. In this study, it was demonstrated for the first time that water-soluble extracts obtained from *Cannabis sativa* L. seeds can inhibit the formation of *S. epidermidis* biofilms. In particular, the extraction process was carried out under different pH conditions. Biocompatible mixtures of fatty acids, sugars, amino acids and glycerol, in different relative amounts as a function of pH, were obtained. Specifically, Ex7d contained a larger variety of sugars compared to the others. Saturated and unsaturated fatty acids were found in all of the analysed extracts, but linoleic acid (C18:2) was detected only in Ex7 and Ex10. All the extracts except for Ex10 exhibited a significant antibiofilm effect against *S. epidermidis*. Ex7d showed the highest antibiofilm activity (i.e., around 90%) and for this reason was further fractionated by HPLC. The antibiofilm activity of the different fractioned fractions was evaluated. The preliminary chemical characterization by 2D-NMR of the most active fraction suggested that bioactivity can be mainly attributed to the presence of glycerolipids, with a possible synergistic contribution of by the other components. The use of aqueous hempseed extracts as antibiofilm agents could represent an attractive alternative due to their lack of toxicity and well-known positive effects on human health.

This paper is dedicated to the memory of Dr. Mario Malinconico.

## Figures and Tables

**Figure 1 ijms-24-16026-f001:**
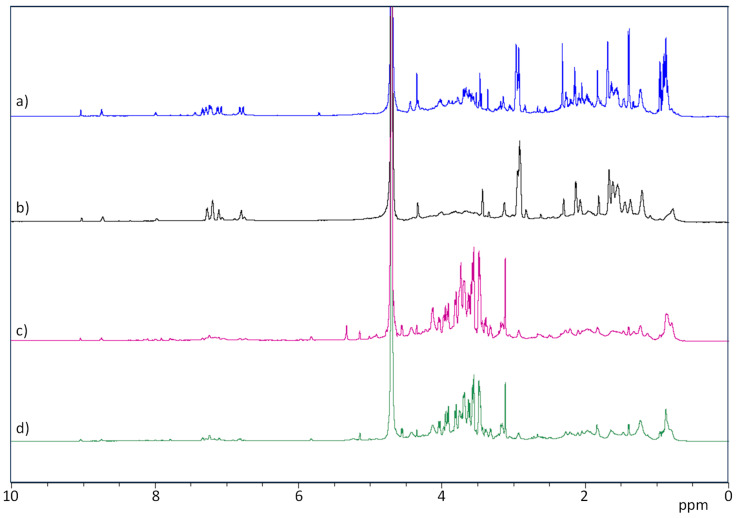
^1^H-NMR spectra of hemp seed Exs. (**a**) Ex7d, (**b**) Ex4.5, (**c**) Ex7 and (**d**) Ex10. Spectra were recorded in D_2_O at 298 K using a 600 MHz instrument.

**Figure 2 ijms-24-16026-f002:**
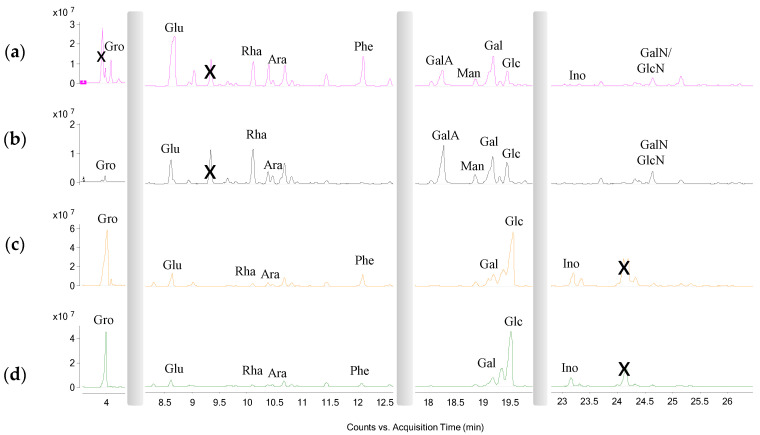
GC-MS chromatogram of AMG and NAME of hempseed Exs. (**a**) Ex7d, (**b**) Ex4.5, (**c**) Ex7, and (**d**) Ex10. Glycerol (Gro), glutamic acid (Glu), rhamnose (Rha), arabinose (Ara), phenylalanine (Phe), galacturonic acid (GalA), mannose (Man), galactose (Gal), glucose (Glc) inositol (Ino), galactosamine (GalN) and glucosamine (GlcN). The peaks marked with X are impurities.

**Figure 3 ijms-24-16026-f003:**
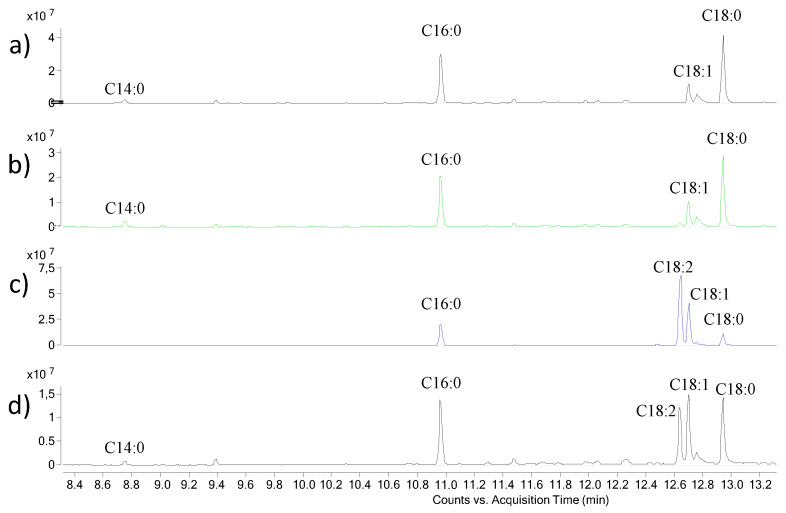
GC-MS chromatogram of FAMEs of hempseed Exs (**a**) Ex7d, (**b**) Ex4.5, (**c**) Ex7 and (**d**) Ex10.

**Figure 4 ijms-24-16026-f004:**
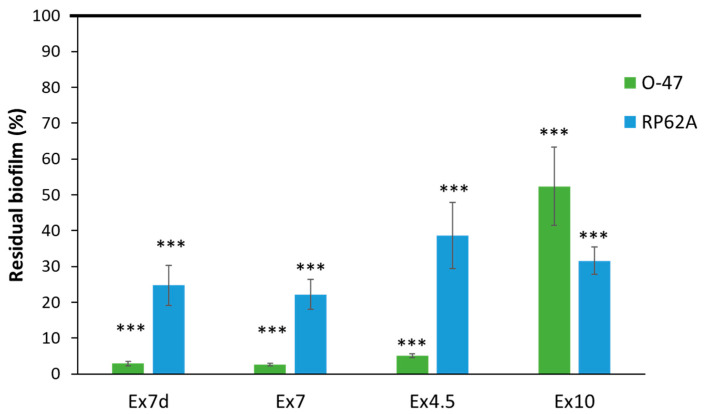
Effect of extracts (1 mg mL^−1^) on biofilm formation by *S. epidermidis* O-47 and *S. epidermidis* RP62A. On the ordinate axis, the percentage of bacterial biofilm production is reported. Data are expressed as the percentage of biofilm formed in the presence of extracts compared with the control sample. Each data point represents the mean ± SD of 5 independent samples. The results are expressed as the percentage of biofilm formed in the presence of the extracts compared to biofilm formed by untreated bacteria (100%). Biofilm formation was considered unaffected in the range of 90–100%. Differences in mean absorbance were compared to the untreated control and considered significant when *p* < 0.05 (*** *p* < 0.001) according to Student’s *t*-test.

**Figure 5 ijms-24-16026-f005:**
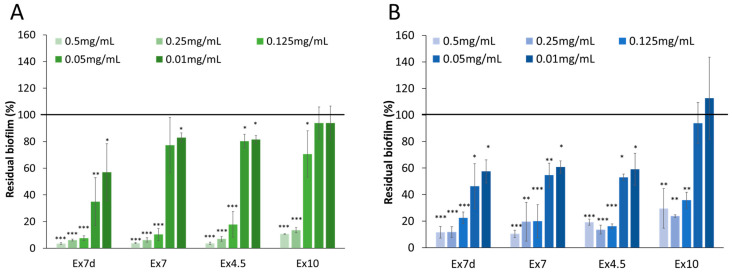
Effect of extracts at different concentrations on biofilm formation by *S. epidermidis* O-47 (**A**) and *S. epidermidis* RP62A (**B**). On the ordinate axis, the percentage of bacterial biofilm production is reported. Data are expressed as the percentage of biofilm formed in the presence of extracts compared with the control sample. Each data point represents the mean ± SD of 3 independent samples. The results are expressed as the percentage of biofilm formed in the presence of the extracts compared to untreated bacteria (100%). Biofilm formation was considered unaffected in the range of 90–100%. Differences in mean absorbance were compared to the untreated control and considered significant when *p* < 0.05 (* *p* < 0.05, ** *p* < 0.01, *** *p* < 0.001) according to Student’s *t*-test.

**Figure 6 ijms-24-16026-f006:**
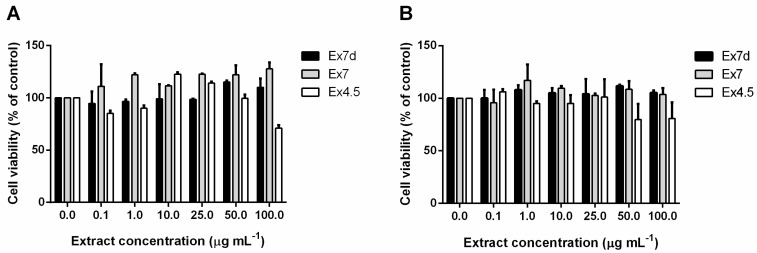
Effect on cell viability of hemp extracts. Dose-response effect of extracts (0.1–100 µg mL^−1^) incubated for 48 h with immortalised human keratinocytes. HaCaT cells were incubated with Ex7d (black bars), Ex7 (grey bars) and Ex4.5 (white bars) for 24 h (**A**) and 48 h (**B**). Cell viability was assessed by the MTT assay and expressed as described in the Materials and Methods section. Values are given as means ± SD (n ≥ 3).

**Figure 7 ijms-24-16026-f007:**
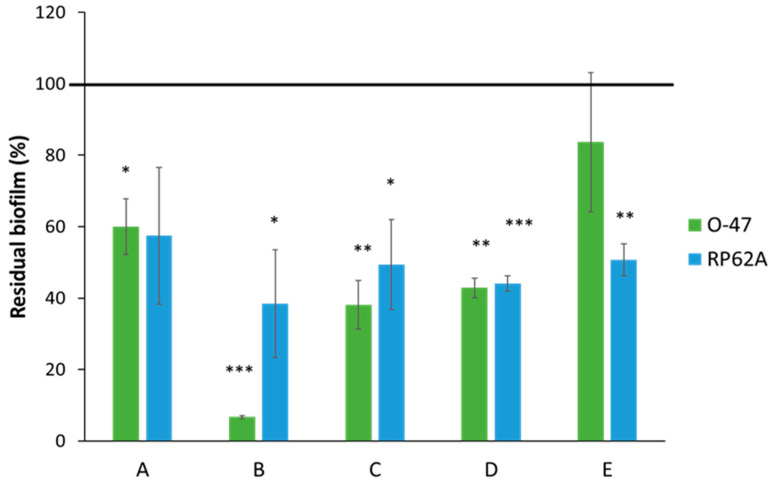
Effect of HPLC fractions (0.1 mg mL^−1^) on biofilm formation by *S. epidermidis* O-47 and *S. epidermidis* RP62A (A–E). On the ordinate axis, the percentage of bacterial biofilm production is reported. Data are expressed as the percentage of biofilm formed in the presence of extracts compared with the control sample. Each data point represents the mean ± SD of 3 independent samples. The results are expressed as the percentage of biofilm formed in the presence of the extracts compared to untreated bacteria (100%). Biofilm formation was considered unaffected in the range of 90–100%. Differences in mean absorbance were compared to the untreated control and considered significant when *p* < 0.05 (* *p* < 0.05, ** *p* < 0.01, *** *p* < 0.001) according to Student’s *t*-test.

**Figure 8 ijms-24-16026-f008:**
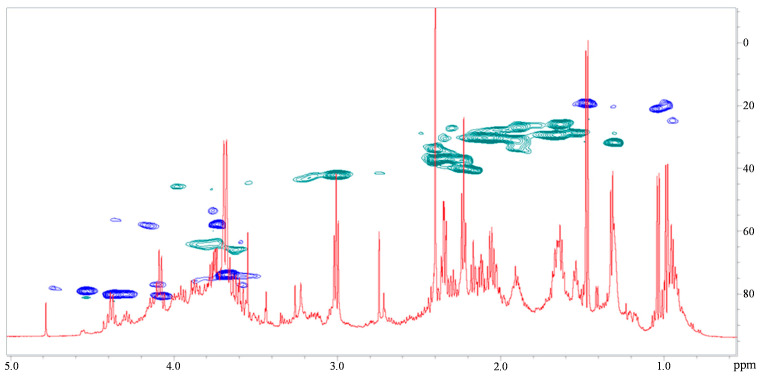
Overlay of ^1^H (red) and 2D ^1^H-^13^C DEPT-HSQC (blue and green) spectra of the B fraction obtained from Ex7b purification by HPLC.

**Table 1 ijms-24-16026-t001:** AMG and NAME retention times and composition (molar %) of hempseed extracts.

	Gro	Glu	Rha	Ara	Phe	GalA	Man	Gal	Glc	Ino	GalN/GlcN
Rt (min)	3.98	8.64	10.11	10.68	12.10	18.26	18.88	19.18	19.52	23.14	24.58/24.61
Ex7d	2.6	26.9	7.6	16.2	10.3	8.0	2.3	17.3	5.5	0.2	3.1
Ex4.5	1.2	7.9	10.6	15.3	-	19.7	3.1	15.5	20.4	-	6.3
Ex7	24.4	4.2	0.7	6.7	4.1	-	-	22.9	31.3	5.7	-
Ex10	21.8	2.6	0.4	6.8	1.5	-	-	22.3	39.5	5.1	-

**Table 2 ijms-24-16026-t002:** FAMEs retention times and composition (molar %) of hempseed extracts.

	C14:0	C16:0	C18:2	C18:1	C18:0
Rt (min)	8.75	10.96	12.64	12.69	12.94
Ex7d	2.7	31.5	-	21.5	44.3
Ex4.5	3.5	30.2	-	25.2	41.1
Ex7	-	13.3	47.4	31.5	7.8
Ex10	1.2	22.1	19.5	32.5	24.7

## Data Availability

Data are contained within the article and Appendix A.

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
