# Peer review of "Aqueous Extracts from Hemp Seeds as a New Weapon against Staphylococcus epidermidis Biofilms"

_ijms, 2023, doi:10.3390/ijms242216026_

Round 1

Reviewer 1 Report

Comments and Suggestions for Authors

Excellent biochemical and analytical work. Here is what I am missing…to declare it as work of immediate medical/health and commercial interest…

There might be significant phenotypic heterogeneity among strains of s. epidermidis response to “Aqueous Extracts From Hemp Seeds” , which is not  addressed either in discussion, or in real experiments. Authors also reported that the value of this variability among 0-47 and RP62V is significant, (see figure 4 and 7) but I did not find elements of discussion related to clinical impact of these diversities in the future directions of “validations”. The first next step should be testing the same extract on the wider strain diversity of S. epidermidis.

Please see and add references and corresponding (self) critical vision in the context of potential applicability.

https://genomebiology.biomedcentral.com/articles/10.1186/gb-2012-13-7-r64

https://www.ncbi.nlm.nih.gov/pmc/articles/PMC6874853/

https://www.sciencedirect.com/science/article/pii/S0944501312001048

Therefore, though I 100% agree with the presented work methods and presented findings, l would be very careful with public enthusiasm, before further microbiological testing is implemented.

Reviewer 2 Report

Comments and Suggestions for Authors

The manuscript “Aqueous Extracts From Hemp Seeds as a New Weapon Against Staphylococcus epidermidis Biofilm” is interesting but not well prepared.

But I have major queries that should be taken into account in the revised manuscript. 

Abstract should include results of GCMS, NMR and HPLC analysis, also mention which extracts have more antibiofilm activity and which chemical constituents responsible for that activity.

Extensive literature needed to include in the introduction part, author performing GCMS, NMR and HPLC analysis no literature reported?

Why author only target carbohydrates, fatty acids and amino acid.

Cannabis sativa contains alkaloids, flavonoids, peptides, tannins, and phenols are also known for antimicrobial activity of Cannabis sativa.

It contains cannbidiol and tetrahydrocannabidiol and many more constituents.

Author should include chemical composition Cannabis sativa using GCMS analysis and which library they used for identification of constituents?

GCMS Method should be in details; not mention run time and chromatographic conditions.

For GCMS they used methanolic extracts and report about sugar in aqueous extracts, same for other extracts; justify?

Method of extraction and sample preparation for fame is not clearly written and no references for all?

Why author perform HPLC?

How you fractionated the extracts?

There is no details about the fraction by using HPLC? Which fraction they used for analysis?

Author provide details results and chromatograms of HPLC analysis?

Also provide GCMS chemical composition of different extract and their chemical constituents with retention time?

How author identified the constituents mention in figure2?

Conclusion should be improved.

Round 2

Reviewer 1 Report

Comments and Suggestions for Authors

Revision is O.K. Good luck.

Author Response

Thank you

Reviewer 2 Report

Comments and Suggestions for Authors

Some of the query significantly responded by authors but most of the queries not significantly responded by authors.

Query 2: Just only one line added by author no extensive literature added in the revised manuscript.

Query 3: Why author only target carbohydrates, fatty acids and amino acid. Cannabis sativa contains alkaloids, flavonoids, peptides, tannins, and phenols are also known for antimicrobial activity of Cannabis sativa. It contains cannbidiol and tetrahydrocannabidiol and many more constituents. Not responded well and not justified authors response this part.

Query 4: Author should include chemical composition Cannabis sativa using GCMS analysis and which library they used for identification of constituents?

Response not justified.

Query 5: For GCMS they used methanolic extracts and report about sugar in aqueous extracts, same for other extracts; justify?

Not justified. why used aqueous extract no explaination.

Query 6: why author perform HPLC?

Not responded well. And chromatogram provided but no results for HPLC analysis. No results about the HPLC peaks and separation.

Why author perform HPLC analysis.

Query : How author identified the constituents mention in Figure 2? R.: The identification of the compounds in the chromatograms was obtained through the interpretation of the mass spectra and the comparison with authentical standards. We usually do not use a library since we possess and use our standards.

Author should provide which standard that used for the identification of constituents?

All the above mention response and query this article not suitable for the publication in IJMS in this current form, therefore it must be reject in the current form.

If author can respond the mention queries and justify all the queries than it may be consider for the publication.

Round 3

Reviewer 2 Report

Comments and Suggestions for Authors

Author significantly revised the manuscript but I have concern about GCMS Supplementary figure S1 a) standard galactose have several peaks in standard chromatogram?

Author Response

We would like to thanks the reviewer for giving us the opportunity to significantly improve our manuscript. We usually prepare standards containing more than one monosaccharide. The standard mixture reported in Figure S1 (a) is only an example and it was prepared by including both galactose and galacturonic acid. We have modified the caption and the text accordingly.